# Sensitivity analysis on the declining population in Japan: Effects of prefecture-specific fertility and interregional migration

**Ryo Oizumi**[1]*, **Hisashi Inaba**[2], **Takenori Takada**[3], **Youichi Enatsu**[4], **Kensaku Kinjo**[5]

**1** National Institute of Population and Social Security Research, Tokyo, Japan, **2** Graduate School of Mathematical Science, The University of Tokyo, Tokyo, Japan, **3** Graduate School of Environmental Earth Science, Hokkaido University, Sapporo, Hokkaido, Japan, **4** Oshamambe Division, Institute of Arts and Sciences, Tokyo University of Science, Tokyo, Japan, **5** Academic Support Center, Kogakuin University, Hachioji-shi, Tokyo, Japan

* ooizumi-ryou@ipss.go.jp

**Data Availability Statement:** All relevant data are within the paper and its Supporting information files.

## Abstract

Japan has been facing a population decline since 2010 due to low birth rates, interregional migration, and regional traits. In this study, we modeled the demographic dynamics of Japan using a transition matrix model. Then, from the mathematical structure of the model, we quantitatively evaluated the domestic factors of population decline. To achieve this, we constructed a multi-regional Leslie matrix model and developed a method for representing the reproductive value and stable age distribution using matrix entries. Our method enabled us to interpret the mathematical indices using the genealogies of the migration history of individuals and their ancestors. Furthermore, by combining our method with sensitivity analysis, we analyzed the effect of region-specific fertility rates and interregional migration rates on the population decline in Japan. We found that the sensitivity of the population growth rate to the migration rate from urban areas with large populations to prefectures with high fertility rates was greatest for people aged under 30. In addition, compared to other areas, the fertility rates of urban areas exhibited higher sensitivity for people aged over 30. Because this feature is robust in comparison with those in 2010 and 2015, it can be said to be a unique structure in Japan in recent years. We also established a method to represent the reproductive value and stable age distribution in an irreducible non-negative matrix population model by using the matrix entries. Furthermore, we show the effects of fertility and migration rates numerically in urban and non-urban areas on the population growth rates for each age group in a society with a declining population.

## Introduction

The *total fertility rate* (TFR) in Japan was below the *replacement-level fertility* (RLF) in 1974 (TFR:2.05, RLF:2.11), and this trend has continued for 45 years (TFR:1.36, RLF:2.07 in 2019) [1]. The census by the Ministry of Internal Affairs and Communications (2015) showed that the total population decline began in 2010 [2]. The low birth rate varies with each geo-political

**Funding:** the Japan Society for the Promotion of Science (JSPS) KAKENHI (Grant number 20K14368) the Ministry of Health, Labour and Welfare (Grant number 20AA2007).

**Competing interests:** The authors have declared that no competing interests exist.

area. The TFR in Tokyo was the lowest (1.15), whereas that in Okinawa was the highest (1.82) in 2019. Thus, the overall TFR of Japan was 1.36 [1]. The latest census (2020) has already been published and indicates continuation of the population decline [3]; however, its publication occurred after the writing of this paper.

If the effects of individual region-specific vital rates, such as age-region-specific fertility, survival, and interregional migration rates, on the total population growth in Japan could be numerically estimated, significant measures against population decline, or to optimally plan for it, could be implemented. Thus, the relationship between these vital rates and population growth rate must be revealed to implement these measures.

For a stable population model, sensitivity analysis is among the most practical existing methods to analyze demographic data [4]. This method can numerically compute the primary differential coefficient of an eigenvalue for each entry in a matrix model, referred to as sensitivity. Initially, this method applied a perturbation method for linear operators, which was primarily developed in physics, to finite matrix operators [5, 6]. Nevertheless, this method has been widely applied not only in physics but also in mathematical demography and ecology because it assists in understanding the environmental effects of population dynamics [7]. For example, a sensitivity representation revealed that a variable environment decreases population growth in Markovian process models [8]. Several organisms have evolved to protect vital parameters with higher sensitivity from variable environments [9], while parameters with negative correlation among others in sensitivity were reported to augment the population growth in variable environments [10, 11]. Moreover, artificial control of vital parameters with higher sensitivity may protect endangered species [12]. Many such applications have been developed in the ecological field [13–18]. Caswell summarizes the methodology [19].

The census report by the Japanese Ministry of Internal Affairs and Communications (2015) includes the cross-regional migration history data for five years age groups [2]. There are 47 prefectures in Japan, and the cross-regional migration history consists of prefectures of residence where each person lived now and five years ago. This information enables us to derive interregional migration rates for each five-year period. Additionally, prefecture-specific fertility rates and life tables were published by the Ministry of Health, Labour, and Welfare. We examine the effect of regional traits on population decline by evaluating them through a sensitivity analysis of an extended Leslie matrix, whose entries incorporate age-prefecture-specific fertility and survival rates containing interregional migration rate. This age-structured multi-variable matrix model is called the *generalized Leslie matrix model* [20] or *age stage-classified model* [19].

Rogers (1975) developed the *generalized Leslie matrix model* as a multi-regional life table [21]. The main focus of these studies was parameter estimation using discrete survey data to approximate continuous mathematical models. However, the mathematical structure of the *generalized Leslie matrix model* has not been completely understood as in the conventional Leslie matrix [22], e.g., the mathematical representations of the reproductive value and stable age distribution. Thus far, the mathematical relationship between these eigenvectors and the elements of the *generalized Leslie matrix model* has not been explicitly shown. Understanding the mathematical structures of these eigenvectors is crucial because they compose sensitivity.

To the best of our knowledge, only a few studies are available on the *generalized Leslie matrix model.* Alternatively, in ecology, research on matrix models, to which *generalized Leslie matrix models* belong, is underway. Currently, the mainstream analytical method in matrix models adopts the *vec-permutation method* [23–25]. This method effectively converts a tensor to a vector, and can handle multistates as a single state. For instance, when the population structure consists of several growth categories with two or more different indicators, such as age and body size, these state transitions are represented by a tensor. The vec-permutation

method can address this type of complicated growth process as a transition in a one-growth category. In contrast, it can simplify and abstract several complicated computations for multi-state matrices without revealing the mathematical relationship between the matrix entries and significant indicators, such as eigenvectors. The vec-permutation method has attracted the attention of ecologists because of its high versatility for various matrix models in numerical calculations.

To reflect the mathematical structure of sensitivity, we have developed representations of eigenvectors and characteristic equations by using the Neumann series [26] of the matrix entries. Our method can interpret these mathematical indicators by using a genealogical tree for the migration histories of individuals and their ancestors, and follows the existing *generalized Leslie matrix model* theory. The life history of each individual is then decomposed into a state transition history as "a path of interregional migration" of an individual and cross-generation. These paths comprise a mean reproductive value across generations and specify the interpretation of the "type-reproduction number," which is another indicator providing a population replacement or net reproduction rate, thereby determining the population growth trend. This method can be used to reconstruct the reproductive value and the stable age distribution by employing entries of the *generalized Leslie matrix model.*

We applied the sensitivity analysis to a *generalized Leslie matrix model* in Japan and found that the influence of variations in birth and migration rates on the population growth rate varies by age. In particular, the influence of migration on the population growth rate is high in those aged under 30. The migration rate from populous urban areas to areas with high fertility rates has the most significant effect. Additionally, the results show that changes in the fertility rate of populous urban areas outweigh the effects of migration of people aged over 30 years on population growth. Because the Japanese government conducts a census every five years, we selected the most recent census (2015) when this study began. By comparing the data from 2010 and 2015, we demonstrate that the sensitivity analysis results from 2015 show a fundamental age structure at each area in the population decline phase.

## Mathematical and numerical methods

### Multi-regional Leslie matrix model and assumptions

In this study, we consider Japan as a country divided into $n$ prefectures. When $\omega > 0$ is the maximum attainable age, we suppose that the total number of female cohorts with age $a$ and living in the $j$-th prefecture at the time unit $t$ is $P_t(a, j)$. A migration rate containing survival rates from the $j$-th prefecture to the $i$-th prefecture in a time unit is $0 \leq k_{ij}(a) \leq 1$ and satisfies

$$\sum_{i=1}^{n} k_{ij}(a) \leq 1.$$

Then, the cohort $P_{t+1}(a + 1, i)$ is the summation of the rates of the cohorts migrating from all prefectures to the $i$-th prefecture at the preceding time $t$ such that

$$P_{t+1}(a + 1, i) = \sum_{j=1}^{n} k_{ij}(a)P_t(a, j). \tag{1}$$

The specific configuration of $k_{ij}(a)$ need not be considered in the theoretical development here and will be discussed later.

The offspring reaching age zero at the next time unit $P_{t+1}(0, i)$ is reproduced by multiplying each cohort $P_t(a, i)$ by the age-prefecture specific fertility $f_{ij}(a)$.

$$P_{t+1}(0, i) = \sum_{a=0}^{\omega}\sum_{j=1}^{n} f_{ij}(a)P_t(a, j). \tag{2}$$

Eqs (1) and (2) can be expressed in matrix form as

$$\mathbf{L} := \begin{pmatrix} \mathbf{F}_0 & \cdots & \mathbf{F}_a & \cdots & \mathbf{F}_\omega \\ \mathbf{K}_0 & \mathbf{O} & \cdots & \cdots & \mathbf{O} \\ \mathbf{O} & \mathbf{K}_1 & \mathbf{O} & \cdots & \mathbf{O} \\ \vdots & \mathbf{O} & \ddots & \ddots & \vdots \\ \mathbf{O} & \cdots & \mathbf{O} & \mathbf{K}_{\omega-1} & \mathbf{O} \end{pmatrix}, \tag{3}$$

where

$$\mathbf{F}_a := (f_{ij}(a))_{1\le i,j\le n}, \quad \mathbf{K}_a := (k_{ij}(a))_{1\le i,j\le n}, \quad \mathbf{O} : n \times n - \text{zero matrix}, \tag{4}$$

which is equivalent to the multi-regional Leslie matrix model given below.

$$\mathbf{p}_{t+1} = \mathbf{L}\mathbf{p}_t, \tag{5}$$

where $\mathbf{p}_t$ represents $n(\omega + 1)$-population vector at time $t$ such that

$$\mathbf{p}_t := (\,\mathbf{p}_t(0) \quad \cdots \quad \mathbf{p}_t(a) \quad \cdots \quad \mathbf{p}_t(\omega)\,)^\top, \quad \mathbf{p}_t(a) := (P_t(a, j))_{1\le j\le n}^\top. \tag{6}$$

Eq (5) belongs to *generalized Leslie matrix models*.

Because the data of the time series related to sufficient prefecture-specific international migration were not obtained, we chose to address the blockade population by eliminating international migration. The inflow of international migration in Japan exceeded 100,000 every five years by an estimation until 2015 [27]. This value indicates that international migration is one of the factors in understanding population trends in Japan. However, there are some difficulties in incorporating international migration into our model. For instance, we do not have data regarding prefecture-specific migration, survival, and fertility rates of foreigners. Some of these data in the Japan census are difficult to divide into residents and immigrants. Consequently, we consider only interregional migration of Japan in this study.

## Eigenvectors and cohorts

We derive the right eigenvector $\mathbf{w}$ of $\mathbf{L}$ corresponding to the eigenvalue $\lambda \neq 0$ from Eq (1) as a step toward the sensitivitystr analysis. Each component $w(a, i)$ of $\mathbf{w}$ should satisfy

$$\lambda w(a + 1, i) = \sum_{j=1}^{n} k_{ij}(a)w(a, j), \tag{7}$$

$$\lambda w(0, i) = \sum_{a=0}^{\omega}\sum_{j=1}^{n} f_{ij}(a)w(a, j). \tag{8}$$

By solving Eq (7) formally, we obtain a component of the right eigenvector:

$$w(a, i) = \sum_{j=1}^{n} K_{ij}(a|0)\lambda^{-a}w(0, j),$$ (9)

where

$$K_{ij}(a|s) := \begin{cases} \sum_{j_1, \cdots j_{a-s-1}=1}^{n} k_{ij_1}(a-1)k_{j_1j_2}(a-2)\cdots k_{j_{a-s-1}j}(s) & s < a-1 \\ \\ \delta_{ij} & s = a-1 \end{cases}.$$ (10)

$\delta_{ij}$ denotes Kronecker's delta. Thus, the right eigenvector consists of the sum of all probabilities of interregional migration paths containing each region-specific survival rate. Because the matrix $\mathbf{L}$ is a non-negative real square matrix, it follows the Perron–Frobenius theorem [28, 29] describing the existence of the dominant eigenvalue $\lambda_1$ that is always positive and real. Let $\mathbf{v}$ be the left eigenvector corresponding to $\lambda \neq 0$. We arrange $\mathbf{w}$ as

$$\mathbf{w} = \begin{pmatrix} \mathbf{w}(0) & \cdots & \mathbf{w}(a) & \cdots & w(\omega) \end{pmatrix}^{\top},$$ (11)

$$\mathbf{w}(a) := \begin{pmatrix} w(a, 1) & \cdots & w(a, i) & \cdots & w(a, n) \end{pmatrix}^{\top}.$$ (12)

Similarly, $\mathbf{v}$ is given by

$$\mathbf{v} := \begin{pmatrix} \mathbf{v}(0) & \cdots & \mathbf{v}(a) & \cdots & \mathbf{v}(\omega) \end{pmatrix},$$ (13)

$$\mathbf{v}(a) := \begin{pmatrix} v(a, 1) & \cdots & v(a, j) & \cdots & v(a, n) \end{pmatrix}.$$ (14)

Suppose that $\mathbf{w}_k$ is the right eigenvector corresponding to the $k$-th eigenvalue $\lambda_k$ ($|\lambda_k| \geq |\lambda_{k+1}|$), $\mathbf{w}_1$ corresponds to the dominant eigenvalue $\lambda_1$ and represents a *multi-regional stable age distribution* (MSAD) for an adequately long time [30]. Focusing on the power term $\lambda_1^{-a}$ in $\mathbf{w}_1$, $\mathbf{w}_1$ is biased toward younger or elderly individuals in the phase of population increase $\lambda_1 \geq 1$ or decrease $\lambda_1 < 1$. In contrast, the left eigenvector $\mathbf{v}_1$ corresponding to the dominant eigenvalue $\lambda_1$ is called the reproductive value, which represents the measure of the contribution of females at each age to population growth.

Substituting Eq (9) in Eq (8), we obtain each component of the right eigenvector at age zero that satisfies

$$\mathbf{w}(0) = \mathbf{\Psi}(\lambda)\mathbf{w}(0),$$ (15)

$$\mathbf{\Psi}(\lambda) := (\psi_{ij}(\lambda))_{1 \leq i,j \leq n},$$ (16)

$$\psi_{ij}(\lambda) := \sum_{a=0}^{\omega}\sum_{\ell=1}^{n}\lambda^{-a-1}f_{i\ell}(a)K_{\ell j}(a|0).$$ (17)

$\mathbf{\Psi}(\lambda)$ is a non-negative matrix. However, we add irreducibility to $\mathbf{\Psi}(\lambda)$ from a practical perspective to implement the Perron–Frobenius theorem to its analysis. An irreducible non-negative matrix to finite power can be a positive matrix, implying that individuals from any birthplace will have descendants in all prefectures in a sufficiently large number of generations. For the solution of Eq (15) to exist, $\lambda \neq 0$ must satisfy

$$\det(\mathbf{I} - \mathbf{\Psi}(\lambda)) = 0,$$ (18)

where **I** represents an $n \times n$ identity matrix. Therefore, Eq (18) is the characteristic equation of Eq (3), and the value of $\lambda \neq 0$ satisfying Eq (18) becomes its eigenvalue. In general, the eigenvalues of Eq (3) include zero; however, this is not considered here because this value does not contribute to population growth. Although all eigenvalues containing zero exist in $\det(\lambda \mathbf{I} - \mathbf{L})$ = 0, all eigenvalues constituting the population growth can be derived from Eq (18) [30]. According to the Perron–Frobenius theorem [29], the dominant eigenvalue $\rho(\lambda)$ of $\Psi(\lambda)$ satisfies

$$\min_i \sum_j \psi_{ij}(\lambda) \leq \rho(\lambda) \leq \max_i \sum_j \psi_{ij}(\lambda). \tag{19}$$

Because $\psi_{ij}(\lambda)$ monotonically decreases in $\lambda$ and the eigenvalue of **L** is $\rho(\lambda) = 1$, the positive real eigenvalue $\lambda_1$, which is the root of Eq (18), is unique. Additionally, $\lambda_1$ is the dominant eigenvalue of **L**.

Moreover, the matrix $\Psi(\lambda)$ provides the eigenvalues and a threshold for population growth. The relationship between $\Psi(1)$ and its dominant eigenvalue $\rho$ is given by

$$\det(\rho \mathbf{I} - \Psi(1)) = 0. \tag{20}$$

The dominant eigenvalue $\lambda_1$ of **L** satisfies

$$\text{sgn}(\rho - 1) = \text{sgn}(\lambda_1 - 1). \tag{21}$$

The function $\text{sgn}(x)$ represents the sign function. Because $\Psi(1)$ matches the requirements of the Perron–Frobenius theorem, $\rho$ is always positive and real. Because this value gives the relative size of the next-generation population, it is equivalent to the average number of reproductions per female. Therefore, $\rho$ denotes the *basic reproduction number* ($R_0$ or *net reproduction rate*) in the *generalized Leslie matrix model* [20].

Similarly, the left eigenvector **v** also plays a crucial role in the sensitivity analysis. Because the left eigenvector corresponds to the transposed matrix $\mathbf{L}^\top$, the transposed equation is given by

$$\lambda v(a,j) = \sum_{i=1}^{n} v(0,i) f_{ij}(a) + \sum_{i=1}^{n} v(a+1,i) k_{ij}(a). \tag{22}$$

By solving the recurrence relation, a component of the left eigenvector is

$$v(a,j) = \sum_{i=1}^{n} v(0,i) \sum_{x=a}^{\omega} \lambda^{-(x-a)-1} \sum_{\ell=1}^{n} f_{i\ell}(x) K_{\ell j}(x|a). \tag{23}$$

From the definition of $\psi_{ij}(\lambda)$, the left eigenvector at age zero $\mathbf{v}(0)$ can be used as a solution to the following equation.

$$\mathbf{v}(0) = \mathbf{v}(0) \Psi(\lambda). \tag{24}$$

Inaba (1986) reported both representations of eigenvectors by using entries of Eq (5) (Eqs (9) and (23)) and verified the existence of solutions in Eqs (15) and (24). Furthermore, Inaba (1986) demonstrated the uniqueness of the multi-regional reproductive value at age zero $\mathbf{v}(0)$, but did not treat the concrete mathematical representation by the entries of $\Psi(\lambda_1)$. We derive the exact forms of both solutions in Eqs (15) and (24) by the entries of Eq (5), as described below.

The conventional interpretation of reproductive value does not include contributions over each generation to reproduction because simple age-structured models have only one state at

age zero. In models with a bias of vital rates for each area, the birthplaces of mothers influence each population structure area. Females born in areas with higher birth rates tend to be the main constituent of the total population. Even in the case of mothers born in areas with low birth rates, their granddaughters contribute to the total population if they move to and give birth to their daughters in an area with a high birth rate. In this case, the contribution of a female born in the $j$-th prefecture to her granddaughters born in the $i$-th prefecture can be represented by a product of two multi-regional reproductive values and the sum over her daughter's birthplace other than the $j$-th one ($\psi_{ij}^{(1)}(\lambda_1)$), as given below.

$$\psi_{ij}^{(1)}(\lambda_1) := \sum_{i_1 \neq j} \psi_{ii_1}(\lambda_1) \psi_{i_1 j}(\lambda_1). \tag{25}$$

By the same token, the degree of contribution after $(m + 1)$-generations is

$$\psi_{ij}^{(m)}(\lambda_1) := \sum_{i_1, \cdots, i_m \neq j}^{n} \psi_{ii_1}(\lambda_1) \psi_{i_1 i_2}(\lambda_1) \cdots \psi_{i_m j}(\lambda_1). \tag{26}$$

The total contribution of a female born in the $j$-th prefecture to descendants in the $i$-th prefecture is represented by adding all contribution degrees from each generation.

$$\psi_{ij}(\lambda_1) + \sum_{m=1}^{\infty} \psi_{ij}^{(m)}(\lambda_1). \tag{27}$$

The series in Eq (27) always converges (Text A in S1 File), and the mean contribution to the same birthplace always converges to one for all $j$ (Text B in S1 File) such that

$$\psi_{jj}(\lambda_1) + \sum_{m=1}^{\infty} \psi_{jj}^{(m)}(\lambda_1) = 1. \tag{28}$$

When the $j$-th component of the right eigenvector is fixed as an arbitrary constant $w_1(0, j) \neq 0$, Eq (27) satisfies Eq (15) (Text A in S1 File). Therefore, Eq (27) configures the components of $\mathbf{w}_1$ at age zero.

$$\mathbf{w}_1(0) = w_1(0, j) \left( \psi_{ij}(\lambda_1) + \sum_{m=1}^{\infty} \psi_{ij}^{(m)}(\lambda_1) \right)^{\top}_{1 \leq i \leq n}. \tag{29}$$

Moreover, because the transposed $\psi_{ij}(\lambda_1)$ generates the reproductive value at age zero (Eq (24)), we can adopt the same method to obtain Eq (29). Subsequently, we focus on a descendant born in the $i$-th prefecture. The contributions of ancestors born in the $j$-th prefecture to the descendant from the $i$-th prefecture are expressed by the generation when descendents appear for the first time. For instance,

$$\bar{\psi}_{ij}^{(m)}(\lambda_1) := \sum_{j_1, \cdots, j_m \neq i}^{n} \psi_{ij_1}(\lambda_1) \psi_{j_1 j_2}(\lambda_1) \cdots \psi_{j_m j}(\lambda_1). \tag{30}$$

Eq (30) implies the contribution degree of an ancestor who has a descendant born in the $i$-th prefecture for the first time after $m$-generation. If we select a descendant born in the $i$-th prefecture, which implies choosing the $i$-th component of the left eigenvector as an arbitrary

constant, the reproductive value at age zero is given by,

$$\mathbf{v}_1(0) = v_1(0,i)\left(\psi_{ij}(\lambda_1) + \sum_{m=1}^{\infty}\bar{\psi}_{ij}^{(m)}(\lambda_1)\right)_{1\le j\le n}, \tag{31}$$

where $v_1(0,i)\neq 0$ denotes an arbitrary constant. $\bar{\psi}_{ij}^{(m)}(\lambda_1)$ has a different value from $\psi_{ij}^{(m)}(\lambda_1)$, but they take the same value if and only if $i$ is equal to $j$, such that

$$\bar{\psi}_{jj}^{(m)}(\lambda_1) = \psi_{jj}^{(m)}(\lambda_1). \tag{32}$$

From the inverse matrix formula, we can verify that each infinite series in the right and left eigenvectors (Eqs (29) and (31)) converges to the following cofactors.

$$w(0,i) = \begin{cases} w(0,j)\det(\mathbf{I}_j - \mathbf{\Psi}_j)^{-1}\Delta_{ij} & j < i \\ w(0,j) & j = i \\ w(0,j)\det(\mathbf{I}_j - \mathbf{\Psi}_j)^{-1}(-1)\Delta_{ij} & j > i \end{cases} \tag{33}$$

$$v(0,j) = \begin{cases} v(0,i)\det(\mathbf{I}_i - \mathbf{\Psi}_i)^{-1}\Delta_{ij} & i < j \\ v(0,i) & i = j \\ v(0,i)\det(\mathbf{I}_i - \mathbf{\Psi}_i)^{-1}(-1)\Delta_{ij} & i > j \end{cases} \tag{34}$$

$\mathbf{I}_\ell$ represents an $(n-1) \times (n-1)$ identity matrix, and $\mathbf{\Psi}_\ell$ represents the $(n-1) \times (n-1)$ matrix, deleting the $\ell$-th row and column of the matrix $\mathbf{\Psi}$. $\Delta_{ij}$ is the $i,j$ cofactor of the matrix $\mathbf{I} - \mathbf{\Psi}$ in Eq (18) (Text B in S1 File). Although these expressions are reasonable for computing the values of eigenvectors, it would be difficult to derive the demographic sense from them.

## Reinterpretation of type-reproduction number from a demographic perspective

Focusing on the following series, comprising Eq (28) as a function $g_j(y)$ with respect to $y > 0$,

$$g_j(y) := \psi_{jj}(y) + \sum_{m=1}^{\infty}\psi_{jj}^{(m)}(y). \tag{35}$$

Suppose that $y_0 > 0$ gives the spectral radius of $\mathbf{\Psi}_j(y)$ as one;

$$\Lambda(\mathbf{\Psi}_j(y_0)) = 1. \tag{36}$$

According to (Text A in S1 File), $y_0$ is always less than the dominant eigenvalue $\lambda_1 > y_0$. For each fixed $y > y_0$, this infinite series converges to the following value.

$$g_j(y) = \psi_{jj}(y) + \mathbf{q}_j(y)(\mathbf{I}_j - \mathbf{\Psi}_j(y))^{-1}\bar{\mathbf{q}}_j(y), \tag{37}$$

where each ingredient represents the row vector $\mathbf{q}_j(y) := (\psi_{ji}(y))_{1\le i\neq j\le n}^{\top}$, the leading submatrix $\mathbf{\Psi}_j(y) := (\psi_{ik}(y))_{1\le i,k\neq j\le n}$, and column vector $\bar{\mathbf{q}}_j(y) := (\psi_{ij}(y))_{1\le i\neq j\le n}$ under a fixed $j$ (see the Theorem proof in Text A of S1 File). By using these properties of $g_j(y)$, we can claim the following theorem (the proof is Text D in S1 File).

**Theorem 1** *Let $g_j(y)$ be given by* Eq (35), *and $\lambda_1 \in \mathbb{R}_+ \backslash \{0\}$ be the dominant eigenvalue satisfying* Eq (18). *Then, $g_j(1) \leq 1$ is equivalent to $\lambda_1 \leq 1$, and $1 < g_j(1) \leq \infty$ in $\lambda_1 > 1$ for all $1 \leq j \leq n$.*

$g_j(1)$ indicates population growth and is identical to the *type-reproduction number* ($\mathcal{T}_j$) defined by Inaba (2009), if it is convergent. The general definition of $\mathcal{T}_j$ is given in Text C in S1 File. The $\mathcal{T}_j$ of the $j$-th prefecture $\mathcal{T}_j$ is defined by substituting $y = 1$ in Eq (37) such that

$$\mathcal{T}_j = \psi_{jj}(1) + \mathbf{q}_j(1)(\mathbf{I}_j - \mathbf{\Psi}_j(1))^{-1}\bar{\mathbf{q}}_j(1). \tag{38}$$

Then, the condition that the spectral radius of $\mathbf{\Psi}_j(1)$ should be less than one (that is, $\Lambda(\mathbf{\Psi}_j(1)) < 1$) is required. Alternatively, we can derive another argument from Theorem 1, which is a reinterpretation of $\mathcal{T}_j$ as the limit of $g_j^M(1)$, paying attention to the existence of $M_0$. Thus, because $\mathcal{T}_j$ is deemed to be the limit of the partial sum,

$$\mathcal{T}_j := \lim_{M \uparrow \infty} g_j^M(1) = \lim_{M \uparrow \infty} \mathcal{T}_j^M, \quad \mathcal{T}_j^M := \psi_{jj}(1) + \mathbf{q}_j(1) \sum_{m=0}^{M} \mathbf{\Psi}_j(1)^m \bar{\mathbf{q}}_j(1). \tag{39}$$

We can define $\mathcal{T}_j$ for all prefectures, regardless of the spectral radius of $\mathbf{\Psi}_j(1)$. Demographically, $M_0$ is the minimum number of intermediate generations in which a female's direct descendants (or ancestors) whose birthplace is the same as hers would be expected to exceed one person. Thus, we can determine the tendency of population growth by the partial sum of $g_j(1)$.

The original definition of $\mathcal{T}_j$ is for a set of regions (prefectures) that are arbitrarily divided into urban and rural areas (Text B in S1 File). The concept of $\mathcal{T}_j$ originates from epidemiology; epidemiologists formulate type reproduction numbers to predict epidemic of an infectious disease from each target condition, such as by classifying infectious individuals into several types [31]. Each $\mathcal{T}_j$ serves as the threshold for a complete system, similar to the $R_0$, although each type has a specific value [32].

Considering the aspects mentioned above, we adopt Eq (39) as $\mathcal{T}_j$. It describes the expectation of recursions to the same birthplace (the $j$-th prefecture in this case) in descendants. In particular, when a population does not increase ($\lambda_1 \leq 1$), $\mathcal{T}_j$ is less than one in all prefectures. Conversely, in the increasing population phase ($\lambda_1 > 1$), all females can find among their ancestors more than one or infinite persons whose birthplace is the same as their own ($\mathcal{T}_j^M > 1$ or $\mathcal{T}_j > 1$).

## Sensitivity analysis

Given the product of the left $v$ and right $w$ eigenvectors corresponding to $\lambda$,

$$\mathbf{vw} := \sum_{a=0}^{\omega} \sum_{i=1}^{n} v(a, i) w(a, i). \tag{40}$$

Each sensitivity $s_{h\ell} := \frac{\partial \lambda}{\partial l_{h\ell}}$ for the general entry $l_{h\ell}$ (containing irrelevant entries for reproduction and migration) in Eq (3),

$$\mathbf{L} = (l_{h\ell})_{1 \leq h, \ell \leq n(\omega+1)} \tag{41}$$

constitutes a sensitivity matrix

$$\mathbf{S} := (s_{h\ell})_{1 \leq h\ell \leq n(\omega+1)} = \frac{\mathbf{v} \otimes \mathbf{w}}{\mathbf{vw}}, \tag{42}$$

where $\otimes$ denotes the Kronecker's product [29]. Each sensitivity is then uniquely determined because the formula for the sensitivity describes the cancelation of arbitrary constant factors in eigenvectors. Although population growth is sensitive to all entries, those with demographic meanings are partially involved in fertilities $f_{ij}(a)$ and migration rates $k_{ij}(a)$ only. Thus, all entries can affect all eigenvalue sensitivities in general. The sensitivity of the dominant eigenvalue $\lambda_1$ has an essential meaning in structured population models. In general, the irreducible non-negative matrix $\Psi(\lambda)$ contains several complex eigenvalues whose absolute values are equal to their spectral radius. However, because the dominant eigenvalue is equivalent to the spectral radius of $L$, calculating the sensitivity of the dominant eigenvalue $\lambda_1$ is sufficient for asymptotic inference.

Focusing on the entries of $\mathbf{S}_1$, which is the sensitivity matrix of $\lambda_1$, each sensitivity of $\lambda_1$ to the fertility follows

$$\frac{\partial \lambda_1}{\partial f_{ij}(a)} = s_{h\ell}\Big|_{h=i,\ell=j+an} = \frac{v_1(0,i)w_1(a,i)}{\mathbf{v}_1\mathbf{w}_1}. \tag{43}$$

Eq (43) implies that the prefecture, where the product of population in $\mathbf{w}_1$ and the reproductive value at age zero is the largest, contributes most to population dynamics. The mathematical structure of sensitivity of $\lambda_1$ to fertility reflects each component $w_1(a,i)$ in $\mathbf{w}_1$. This structure is identical to that of the Leslie matrix model [4]. Nevertheless, there is a significant point that multi-regional reproductive values at age zero yield regional differences in sensitivity. Moreover, we can find concave/convex-shaped irregularities in $\mathbf{w}_1$ due to interregional migration.

However, the sensitivity of $\lambda_1$ to the interregional migration rate at age $a$ becomes

$$\frac{\partial \lambda_1}{\partial k_{ij}(a)} = s_{h\ell}\Big|_{h=i+(a+1)n,\ell=j+an} = \frac{v_1(a+1,i)w_1(a,j)}{\mathbf{v}_1\mathbf{w}_1}. \tag{44}$$

Eq (44) implies that transitions from prefectures with larger components in $\mathbf{w}_1$ to those with higher reproductive values at the following age contribute to the population dynamics more than other interregional migration. Because of the TFR definition that the birth action occurs at the reproductive age, it categorizes newborns from females in the age group of 15 to 49 in Japanese government statistics. Therefore, all reproductive values of individuals over 50 years old are inevitably zero, and the interregional migration of females aged over 50 years is irrelevant to reproduction.

The sensitivities of the dominant eigenvalue $\lambda_1$ to fertility and interregional migration rates are known. Because the dominant eigenvalue is less than one ($\lambda_1 < 1$) in nations with a declining birth rate, the formula of the right eigenvector (Eq (9)) expects population growth rate to be more sensitive to elder fertility. However, the high frequency of migration from rural to urban areas may differ between these areas for sensitivities at younger ages. Moreover, real data must be input to the model to evaluate these qualitative predictions. The next section describes our construction of a dataset comprising the developed model using Japanese demographic data.

## Parameters and methods for numerical analysis

Several government statistics must be used to achieve our purpose, which is constructing the *generalized Leslie matrix model* for sensitivity analysis of prefecture-specific fertility and interregional migration. The frequencies and age intervals used in age categorization do not always match each other in these statistics. For instance, the Japanese census publishes interregional movement data at each age, but the survey is conducted every five years only. Therefore, the

time unit and age must be aligned. Each age is modified to a five-year age group, and we employ five years as a time unit for simplicity. The demography of Japan defines newborns in the period from October 1 to September 30 of the next year as the cohort of each year.

**Construction of data sets for the model.** To maintain the independence of the original data, we assume that all demographic phenomena occur in a discrete-time unit. Then, birth, death, and migration are assumed to occur every five years when the census is conducted. This assumption is the basis of the regional population projections for Japan [33]. No existing report indicates that this assumption is inconsistent with actual population trends. Therefore, we prioritize the simpler parameters as an appropriate discretization.

Suppose that this migration rate can be separated from the pure migration probability $T_{ij}(a)$ and the survival rate $p_j(a)$

$$k_{ij}(a) = T_{ij}(a) \times p_j(a) \quad i, j = 1, 2, 3, \cdots, n. \tag{45}$$

Here, we adopt survival rates for prefectures where cohorts live at an earlier time step.

First, we derive the transition probabilities from each prefecture for five years. The definition of each transition probability is given by

$$T_{ij}(a-5 \sim a-1) := \frac{\sum_{x=a}^{a+4} P_{2015}(x, j \to i)}{\sum_{x=a-5}^{a-1} P_{2010}(x, j)} \quad a = 5, 10, \ldots, 85. \tag{46}$$

The numerator on the right-hand side represents the total population moving from the $j$-th to $i$-th prefectures in age group $a - 5$ through $a - 1$ until 2015. The remaining probabilities for each prefecture become

$$T_{jj}(a-5 \sim a-1) = 1 - \sum_{i \neq j} T_{ij}(a-5 \sim a-1). \tag{47}$$

The census on the statistics of migration treats populations aged over 85 years as one age group, such as $85_+$ [2]. We set the transition probability of the last age group as

$$T_{ij}(85_+) = \frac{\sum_{x=85}^{105_+} P_{2015}(a, j \to i)}{\sum_{a=80}^{105_+} P_{2010}(a, j)}. \tag{48}$$

According to our framework, Eq (3), this age group is ignored for the following five years. This implies that Japanese females over the age of 90 are not considered, as they do not have an effect on collective reproduction rates in Japan. Because we aim to analyze the effect of regional differences and interregional migration on declining birth, the behavior of the aging population can be excluded for several reasons. First, age groups that are over 50 do not contribute to the value of λ (see the previous section). Second, considering the reproduction of females aged over 85 years would be a mistake from a physiological standpoint.

We computed survival rates in five-year age groups in each prefecture by using a prefecture-specific life table in the vital statistics.

$$p_j(a) = \frac{l_{a+4,j}}{l_{a,j}}, \quad l_{0,j} = 1, \tag{49}$$

where $l_{a,j}$ denotes the survivorship in the $j$-th prefecture. Because human reproduction occurs with continuous migration, females have fertility rates for all prefectures in general, such as $f_{ij}(a)$. However, we have no consistent data supporting the fertility rate incorporating migration. Every five years, the Japanese census inquires the survey participants regarding their residential districts five years prior. Although it publishes data on the migration history of

participants whose age is less than five, they do not match the definition of internal migration at other ages. We assume that newborns inherit the birthplace from where their mothers last lived to resolve these data discrepancies ($f_i(a)$: = $f_{ij}(a)\delta_{ij}$). The fertility rate considered in our system requires various demographic data, such as the sex ratio at birth, infant mortality rate, and birth rate in each prefecture.

$$
f_i(a) \quad = \underbrace{\frac{100}{100 + \sum_{\tau=2011}^{2015} \mathrm{MSB}_\tau / 5}}_{\text{Sex ratio at birth}}
$$
$$
\times \underbrace{\frac{\left(1 - \frac{IFM_i(2015)}{1000}\right) \sum_{k=0}^{4} B_{i,a+k}(2015-k) l_{k,i}(2015)}{1000}}_{\text{Expectation of fertility rate}}. \tag{50}
$$

This fertility rate at age $a$ in the $i$-th prefecture consists of the expectation of the sex ratio at birth and the fertility rate for both sexes [1]. The former incorporates the mean male sex ratio at birth $\mathrm{MSB}_\tau$ for five years at time $\tau$ [1]. The latter comprises infant survivorship in 2015, $\left(1 - \frac{IFM_i(2015)}{1000}\right)$, and the mean female birth rate for five years from 2011 to 2015 [1]. We adopt the national average value in vital statistics to the male sex ratio at birth for simplicity, and the infant mortality $\frac{IFM_i(2015)}{1000}$ and each survivorship cite the life table in 2015 [34]. The birth rate $B_{i,a}(t)$ refers to newborns per 1000 females at age $a$ for a year [1].

**Method for numerical analysis.** There are 47 prefectures and we set 18 age groups in Japan. Accordingly, the computation of an $846 \times 846$ matrix was required. The developed methodology, relying on infinite series to represent eigenvectors and $\mathcal{T}_j$ (Eqs (29), (31), and (39)), is reasonable for interpreting them. However, it is not appropriate for their numerical computation because this type of series always possesses limitations regarding the convergence time. Therefore, we used Mathematica (Wolfram), a commercially available numerical analysis software, to perform direct calculations. Regarding $\mathcal{T}_j$, because $R_0$ in Japan is computed to be less than one ($\rho = 0.69$), we used the conventional definition of Eq (38).

## Results

Herein, we present the simulation results of the proposed models. The dominant eigenvalue $\lambda_1$ from 2015 was approximately 0.94, indicating that the Japanese population was in a decreasing phase, which matches the Japanese census reports [2]. In other words, we accurately simulated the state of Japanese demographics for 2015. Our model calculated this value as smaller than the population statistics because the mean fertility rate contained lower fertility from 2011 to 2015 than in 2015.

### Features of some prefectures in parameters derived from the national statistics

According to vital Statistics (or Population Statistics of IPSS [1, 35]), all prefectures had lower TFR than replacement-level fertility; however, Okinawa demonstrated the highest TFR in recent decades (for example, 1.89 in 2018). In Japan, a higher TFR tends to be concentrated on the southwest region, such as Kyushu. Conversely, the northeastern region (Tohoku region) and Hokkaido have a lower TFR than the other regions. Tokyo has had the lowest TFR for a long time, and a TFR of 1.20 in 2018. The developed model Eq (5) also describes these features;

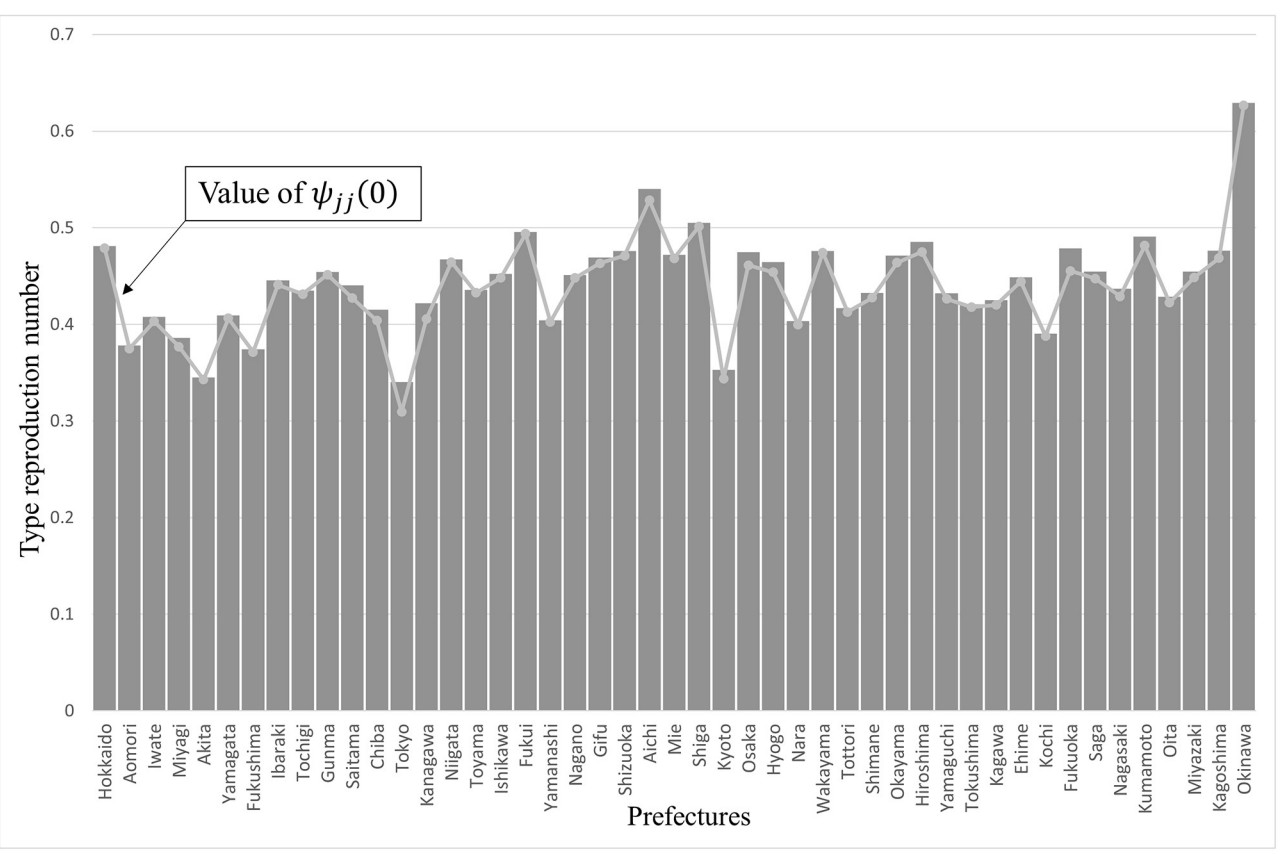

**Fig 1. Type reproduction number in Japan.**

for example, Okinawa, which has *prefecture code* 47 (pc. 47) has the highest fertility.

$$f_{47}(15) + f_{47}(20) + \cdots + f_{47}(50) = 0.92. \tag{51}$$

Incidentally, Eq (51) in Tokyo (pc. 13) is 0.55, which is the smallest value. By comparing these two prefectures, the features appear in each $\mathcal{T}_j$.

Fig 1. shows the TRN for each prefecture and the diagonal components of the $\Psi$. The bars and the line graphs represent the TRN $\mathcal{T}_j$ and the value of $\psi_{jj}(0)$ for each prefecture, respectively. Each $\mathcal{T}_j$ follows the trend of $\psi_{jj}(0)$, which represents a woman reproducing in the same birth place.

In Okinawa, $\mathcal{T}_j$ was approximately 0.63, which is the largest value among all prefectures. In contrast, in Tokyo, the value was nearly 0.34, which was also the smallest value. This discrepancy between the TFR and $\mathcal{T}_j$ occurred because females do not always give birth at their birthplaces. This result suggests that measures against the declining birth rate are not the only issue of fertility, as $\mathcal{T}_j$ must exceed one to stem the population decline. Comparing $\mathcal{T}_j$ of all prefectures $j = 1, \cdots, 47$ contributes to understanding the present prefectures [36]. The TRN structure is a byproduct of the analysis of the eigensystem structure.

Moreover, $R_0$ ($\rho$) in our system Eq (5) is 0.69, which is almost the same as the mean value between 0.68 in 2010 and 0.70 in 2015 [1, 35]. This result implies that the difference between

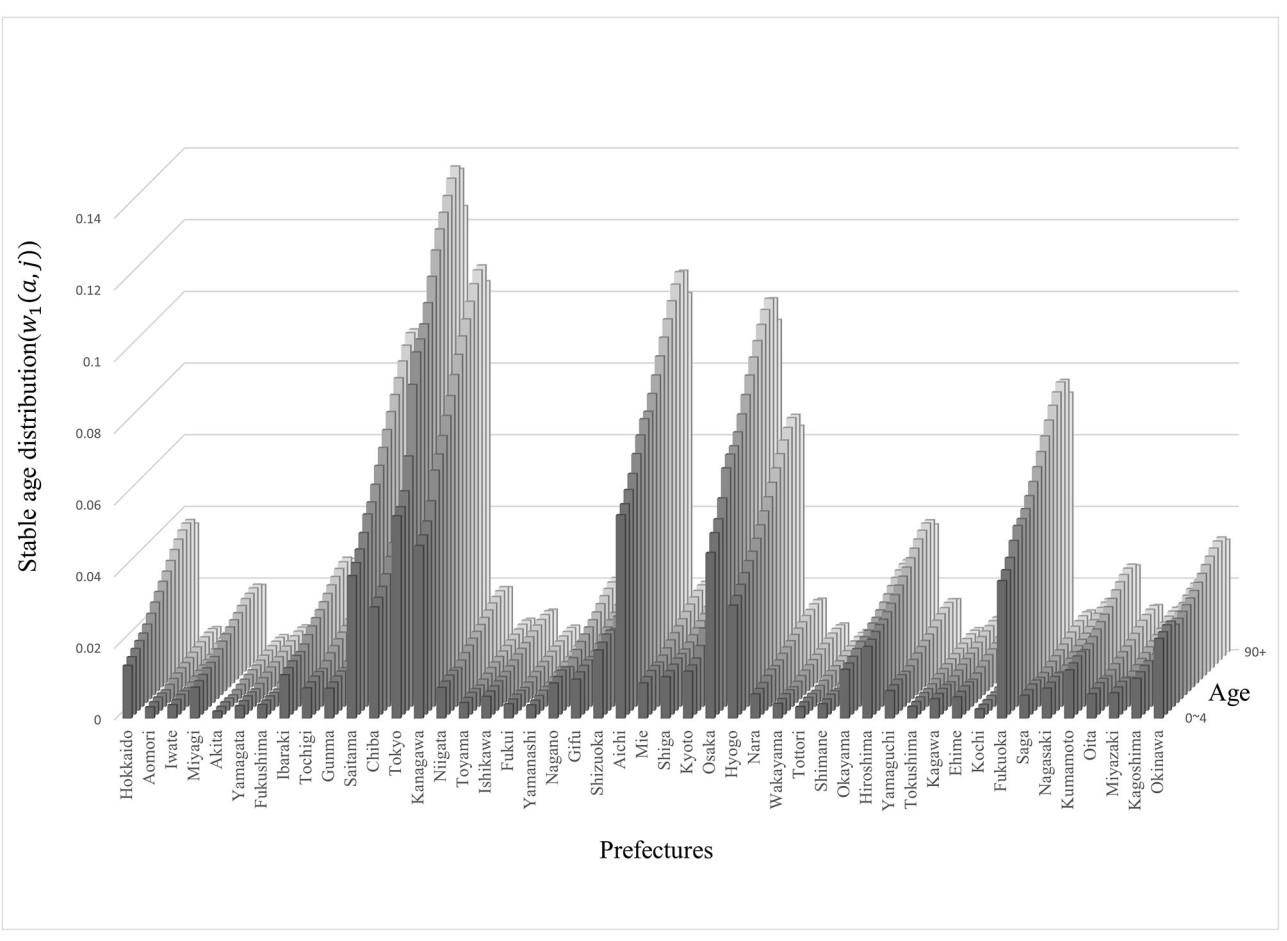

**Fig 2. Multi-regional stable age distribution.**

the actual value and the estimation by our model was small despite the combination of census and demographic statistics.

## Main results

By computing $\mathbf{w}_1$, prefectures with a larger population were found to be concentrated on existing metropolitan areas such as Tokyo, Kanagawa, Aichi, and Osaka (see Fig 2).

The stable age distribution for each prefecture as computed by the government statistics in 2015 is shown. The urban area continues to contain a larger population than the rural area because the population inflow tends to exceed the outflow. Notably, the urban area has a large population volume in the 20s; conversely, the rural area has a unique concave irregularity in that age group.

These prefectures are believed to exhibit inflow excess. The reproductive value depends more on the fertility rate of each prefecture than on the size and migration among areas (Fig 3). For example, Okinawa has the largest reproductive value in the entire age group, despite not having a large population in $\mathbf{w}_1$. Moreover, larger reproductive values are concentrated in southern Japan, where fertility tends to be high.

The left eigenvector corresponding to the dominant eigenvector (reproductive value) computed by the government statistics in 2015 is shown. The southern prefectures tend to have

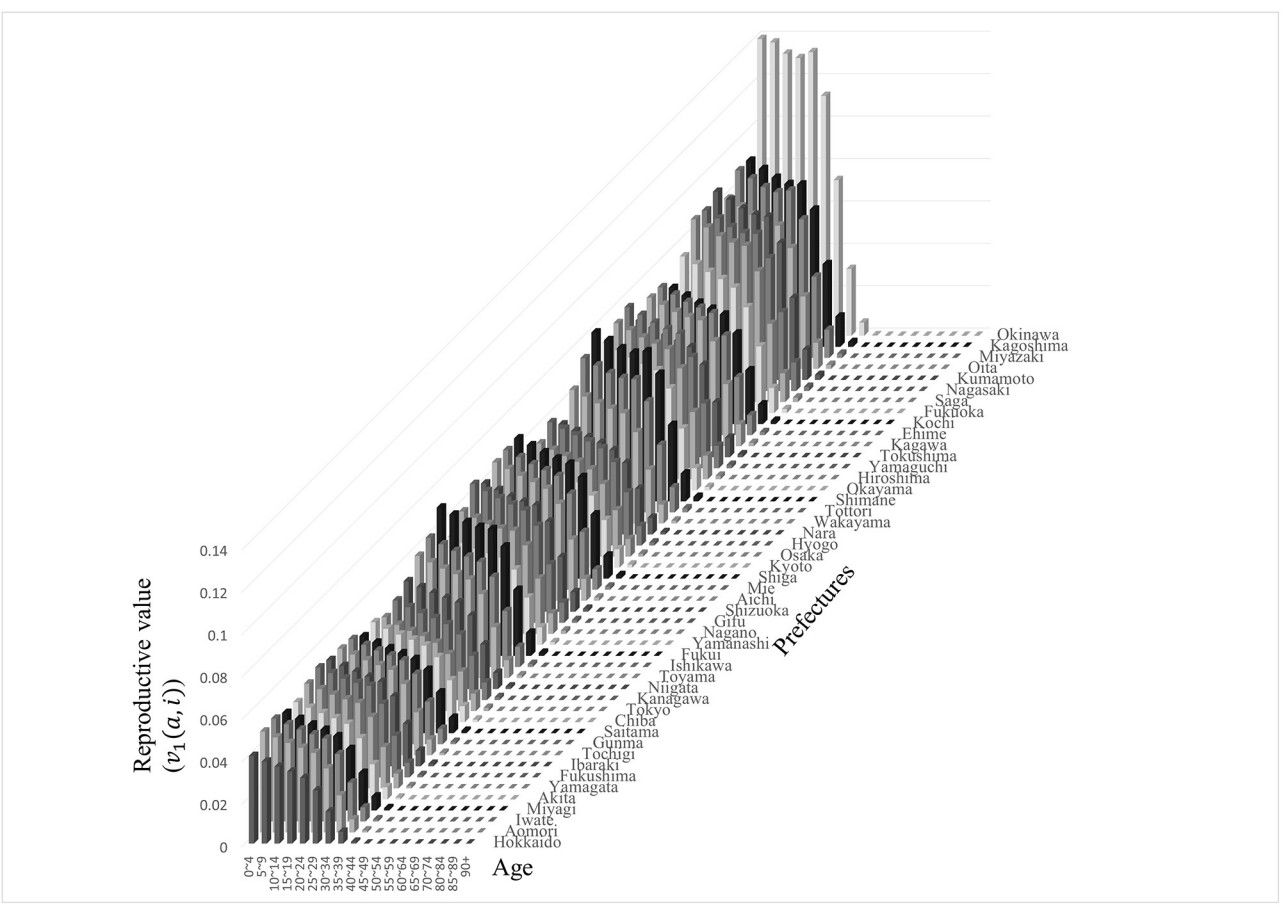

**Fig 3. Multi-regional reproductive value.**

larger reproductive values than other prefectures. Okinawa has the largest reproductive value in Japan. Urban areas such as Tokyo, Kanagawa, Aichi, and Osaka have relatively smaller values.

The mathematical analysis in the previous section reveals that migration rates from any prefecture to Okinawa, which had a large reproductive value for all ages, had the highest sensitivity (Eq (44)). Incidentally, for Okinawa, the anchoring rates had the highest sensitivity. The sensitivity of the population growth to fertility was the product of reproductive values at age zero and $\mathbf{w}_1$, so municipalities with large reproductive values and urban areas with high population density had high sensitivity. Indeed, the top six prefectures with high sensitivity are Tokyo, Kanagawa, Aichi, Osaka, Fukuoka, and Okinawa (Fig 4a and 4b). Fig 1 shows that the composition of prefectures with higher sensitivities does not change significantly compared with that in 2010. In a society with a declining population, the frequency of migration from large population areas in the stable age distribution to areas with large reproductive values may have a significant effect on population growth. In Japan, this result coincides with the frequency of migration from urban areas such as Tokyo to areas with high fertility rates such as Okinawa, affecting population decline. Prefectures that currently host large cities do not always have large populations in the stable age distribution. Miyagi (2.3 million) is classified as an urban area with a larger population than Okinawa (1.4 million) as of 2015 [1], but its population in $\mathbf{w}_1$ is below that of Okinawa (see Fig 2).

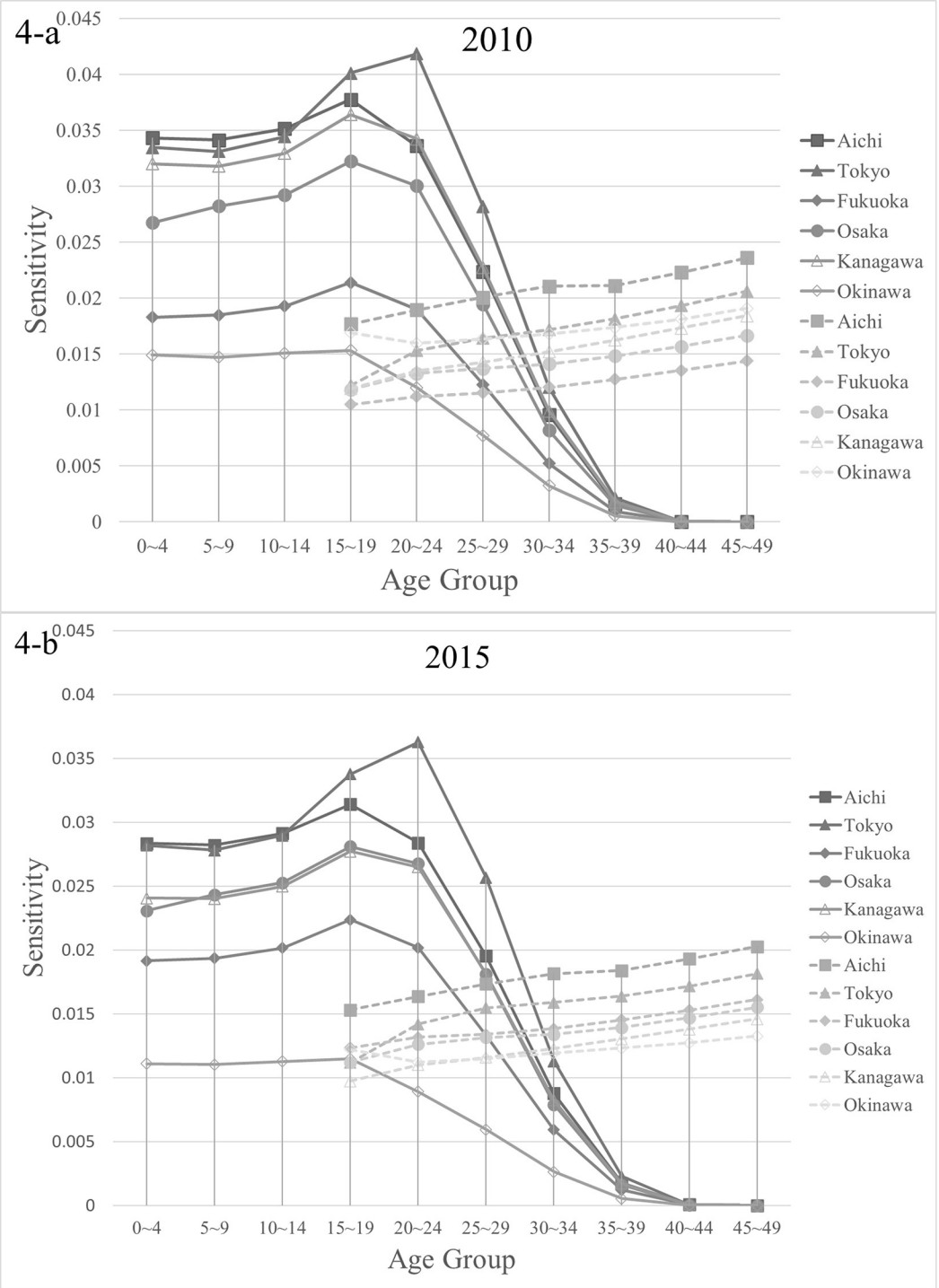

**Fig 4. Comparison with sensitivity in 2010.**

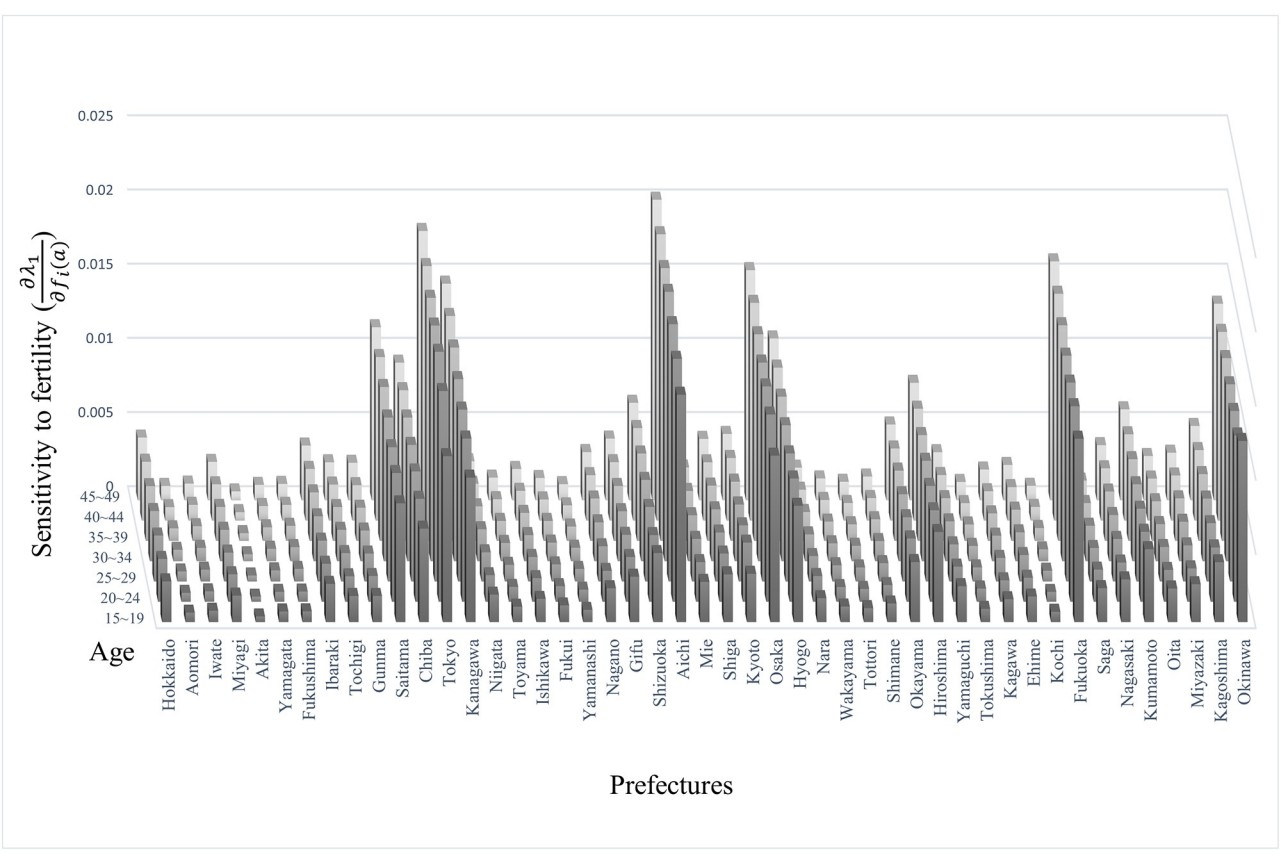

**Fig 5. Sensitivity to fertility.**

Here, the sensitivities in 2015 and 2010 are compared. Fig 4 show the sensitivity of $\lambda_1$ to the interregional migration rates from six prefectures to Okinawa (solid lines) and the fertility of each (dashed lines) in 2010 and 2015. These six prefectures had significantly higher sensitivity than other prefectures. The sensitivity of $\lambda_1$ to flow to Okinawa became the highest in all interregional migration in Japan because Okinawa had the largest reproductive values (cf. Fig 3). The trends of the sensitivity structure remain almost unchanged for the recent five years. These figures show that the replacement age from the effect of migration to fertility on the population growth did not change.

The critical point here is that the sensitivity of $\lambda_1$ to fertility exceeded that of migration rates from age 30. There were some exceptions here (Fig 5); prefectures with metropolitan cities, such as Hokkaido and Miyagi, where Sapporo and Sendai are located, have low sensitivity of $\lambda_1$ to fertility because of their low reproductive value (Fig 3).

This figure illustrates the sensitivity of $\lambda_1$ to fertility for the five-year age groups and each prefecture. The vertical axis represents the five-year age groups, and the horizontal axis represents prefectures in Japan. Eq (43) gives the formula for sensitivity. The structure of the sensitivity reflects the value of $W_1$. Contrastingly, it does not entirely correspond to the distribution because of the difference in values for each reproductive value component at age zero. For example, Okinawa had higher sensitivity than Hokkaido and Miyagi, which contain the urban cities Sapporo and Sendai, but it does not have a larger population.

Conversely, Okinawa does not have metropolitan cities, but its sensitivity was high because of its large reproductive value at age zero. This result indicates that the influence of migration and fertility on population growth differs distinctly between age groups. The reproductive value decreases after the age at which the birth rate falls, but in a society with a declining birth rate, the population increases with age. This structural difference is the cause of the reversal of sensitivity. This suggests that improving fertility decline after the 30s is more effective as a policy for declining birth rate than other age groups.

## Discussion

We have succeeded in linking traditional demographic indicators as statistics of migration history in two different time scales. One represents the migration pathway in each cohort. The probability of cohort migration history with survival explicitly composes the right and left eigenvectors, except for age zero components (Eqs (9) and (23)). This probability depends on the timescale of the lifespan. The other indicator represents the lineage of migration history through ancestors or descendants, composing the regional population distribution at age zero (Eq (29)), regional reproductive value at age zero (Eq (31)), and the type-reproduction number (Eq (39)). This implies that we should consider information on the generation time scale to address statistics for the neonatal state and future reproduction.

The genealogical tree for interregional migration in demography has limited $\mathcal{T}_j$ interpretation to cases where the spectral radius of the transition matrix within the non-target area is less than one [37]. Our proof for the series representation of eigenvector (cf. Text A in S1 File) indicates that the interpretation is generalized to multi-regional population models or more global irreducible non-negative matrix models. Through this interpretation, $\mathcal{T}_j$ need not be a finite value, unlike $R_0$. Even if $\mathcal{T}_j$ becomes infinite, the population growth rate is always finite and is greater than one.

Moreover, the genealogical tree for interregional migration generalizes the Euler–Lotka equation, which provides the eigenvalues of the Leslie matrix. The function $g(\cdot)$ is not only another form of the characteristic polynomial but also shows that the dominant eigenvalue contains information on all recurrent migration pathways of the descendants and ancestors. As for the Perron–Frobenius theorem, all components of eigenvectors corresponding to the dominant eigenvalue are positive, which implies that some migration pathways over generations can connect all prefectures. The theorem we found (Text A and B in S1 File) applies to all irreducible non-negative matrices. In previous analyses of mega-matrices used in ecology [38, 39], eigenvectors, like eigenvalues, were constructed by numerical approximations. Although these values that constitute eigenvectors are functions of matrix entries, the Leslie matrix is the only example in which they express relations that have biological significance. Our analysis is not a life history analysis based on sensitivity or elasticity using the values constituting the eigensystems but an expression of eigenvectors based on matrix entries that give biological meaning to the eigensystems themselves.

The genealogical analysis of population dynamics is applied not only to the analysis of the developed models but also to the field of cell engineering. These studies calculate the population growth rate mainly by examining the genetic lines that frequently appear in a small number of populations [40]. Although these studies do not directly calculate $\mathcal{T}_j$, our method can be applied to them because the matrix entries $\psi_{ij}(1)$ can directly compose the system's statistics. For instance, the representation of eigenvectors using the Neumann series expansion (Text A in S1 File) clearly describes $\mathbf{w}_1$ that reaches across generations. Additionally, this structure quantitatively explains the details of the demographic problems that we currently face, such as sensitivity. We demonstrated that the type reproduction number and a component of the

eigenvector corresponding to $\lambda_1$ can be expressed as a single function via $\lambda$ (see Eq (35) and Theorem 1). In other words, our theorem extends the relationship between the basic reproduction number and the Euler–Lotka equation in the Leslie matrix to irreducible mega-matrices.

Regarding the present data analysis in Japan, the declining birth rate is reproduced in our model, with the values of $R_0$ and $\mathcal{T}_j$ being smaller than one. Therefore, $\mathbf{w}_1$ is biased toward older adults overall, and urban areas with high flows especially have a large elderly population; for example, Tokyo, Kanagawa, Aichi, Osaka, and Fukuoka. Migration is characterized by a high population density in the early twenties in urban areas and low in rural areas. The reason is that this age group migrates from rural areas to urban areas for school or employment, as mentioned before.

Conversely, reproductive values tend to be smaller in urban areas, but larger in areas located geographically to the south. Okinawa has the largest reproductive value among all prefectures of all ages, owing to their having the highest fertility rate. Combined with the characteristics of the stable age distribution and reproductive value, the sensitivity of population growth rate to migration showed that the change in migration rate from urban areas (for example, Tokyo and Aichi) to rural areas (for example, Okinawa) at a younger age had the most effect on the declining birth rate. However, the effect of migration changed on population decline occurred until the age of the 20s in all prefectures. The reason is that the reproductive value dropped sharply after the 30s and became almost zero from the mid-40s.

There are different interpretations from other ages for reproductive value and $\mathbf{w}_1$ at age zero. The former represents the contributions of migration histories by ancestors to descendants from a fixed birthplace. Okinawa, with the largest reproductive value at age zero, had the highest contribution to descendants from all birthplaces. Because the latter had a fixed birthplace of ancestors, Tokyo, with the largest population in $\mathbf{w}_1$ at age zero, implies that it had the highest contribution from other prefectures.

Furthermore, the reproductive value at age zero plays a vital role in multi-regional population models because the sensitivity of the population growth rate to fertility does not depend solely on the stable age distribution. Because the sensitivity of the population growth rate to each prefecture's fertility involves multiplying the prefecture-specific reproductive value at age zero and the stable age distribution, the right eigenvector structure corresponding to the dominant eigenvalue reflects its value. Thus, the fertility of older adults in urban areas affects the declining birth rate society.

However, there are some exceptions: rural areas with large reproductive values and urban areas with small reproductive values at age zero. For example, fertility sensitivity was high even in Okinawa, which was not classified as an urban area. Conversely, Hokkaido and Miyagi, where Sapporo and Sendai are located, had small sensitivity values, even though they were categorized as urban because of their small reproductive value. This characteristic is not observed in age-structured models, such as the Leslie matrix, because they have only one state of reproductive value at each age. The reproductive value at age zero in the age structure model with only one state is equal to an arbitrary constant depending on the property of the eigenvector.

Whereas the effect of migration on population growth decreases after age 30, the sensitivity of the population growth rate to fertility increases with age. Compared with the 2010 data, it may be observed that the sensitivity structure has only slightly changed. Therefore, it is possible to consider policies based on this structure. Thus, this structure could provide one of the population policy considerations.

One of the most significant features of the conducted sensitivity analysis is comparing the effects of interregional migration rates and each prefecture's fertility on population growth by age. Although these parameters have different ranges, they can be compared because the

matrix entries are dimensionless. The results show that the sensitivity is reversed at the age of 30 years. Up to the 20s with large reproductive values, the migration rate from urban areas to prefectures with high birth rates, such as Okinawa, plays a vital role in population growth.

Alternatively, from the 30s onwards, when the reproductive value declines sharply, changes in the birth rate in urban areas with high population density contribute significantly to population growth. This suggests that the structure created by the decline in Japan's birth rate, late marriage, and over-concentration of the population in urban areas increases the sensitivity of the population growth rate to those age groups. Consequently, the increase in the elderly population ratio due to a declining birth rate has a more significant effect on the total number of births than in the younger generation.

The results show that promoting relocation to areas with high birth rates, such as Okinawa, up to the 20s and increasing the birth rate after the 30s is effective in the control of birth rate decline. Because this feature is robust in comparison with that in 2010 and 2015, it can be said to be a unique structure to Japan in recent years. This type of measure could contradict the fundamental theorem for the optimal life schedule [41–45]. In brief, this theorem states that precocious, prolific, and long reproductive ages maximize the intrinsic rate of natural increase, which is correct in the long term. Moreover, it is required to maintain the scale of the population eventually. In contrast, the developed model presupposes population decline and indicates an effective short-term strategy because it arises from the partial derivative around the current data.

Eq (5) assumes that migrants exhibit the same reproductive and death behaviors as people living in each destination prefecture. In reality, however, the life course depends on the length of residence, age, and cultural background of the migrants. Eq (5) cannot treat these traits of migrant cohorts. Furthermore, Japanese governmental statistics do not separate vital statistics between migrants and residents. Even with the large influx of migrants, there is no strong reason for maintaining region-specific vital statistics. In addition, it will be necessary to consider the trend of foreign residents and overseas Japanese because the international population migration is ignored in our model. However, the number of foreign residents up to 2015 was 2,232,189, which is approximately 2.0% of the total population of Japan [1]. Although the number of foreign residents has been increasing at an average of 60,000 per year since 2000, the Great East Japan Earthquake in 2011 and the Fukushima nuclear power plant accident caused a temporary outflow of about 50,000 people per year that continued for approximately three years [1]. However, the sensitivity analysis results show that there was almost no structural change in sensitivity (see Fig 4a and 4b) between 2010 and 2015 after the earthquake, suggesting that the impact of international migration is limited.

It is even more significant to study the regional culture, history, and efforts of each local government that shape the region-specific life course in understanding population decline. Our study points out concrete elements that directly affect the population decline, such as age group, region-specific fertility, and interregional migration rates. Our results are expected to guide studies on the sociological background of the main prefectures that influence population dynamics. Further analysis of sociological factors based on sensitivity analysis is expected to contribute to effective policy-making to counter the declining birth rate. Furthermore, considering that data from Japanese government statistics do not distinguish the duration of residence or nationality, it will be necessary to understand the regional characteristics of fertility and mortality rates based on the ratio of long-term residents to new residents in each municipality to make policy decisions.

Commonly, various developed countries face difficulties in realizing fundamental theorems. Several social issues involved in the decline of birth rate, such as gender gap, education, and wealth inequality, must be addressed. However, our sensitivity analysis suggests that a

short-term strategy may be more effectual in buffering the population decline than long-term measures. Therefore, in addition to immigration measures to solve the problem of birth rate decline, we propose considering the accumulation of short-term strategies as a viable method.

## Supporting information

**S1 File. Dummy.** Text A, Representation theorem for a right eigenvector of an irreducible non-negative matrix. Text B, Theorem for infinite series expansion of characteristic equation. Text C, Original definition of type-reproduction number. Text D, Extension theorem of type-reproduction number.
(ZIP)

## Acknowledgments

We are deeply grateful to Shiro Koike for his helpful comments and advice on this study. Further, we thank Shohei Yoda, Kota Hattori, and Nobuhiko Fujii for their helpful comments and advice on this study, and Kumiko Oizumi, Kunihiro Aoki, Hiroko Oizumi, Setsuya Fukuda, and Ryuichi Kaneko for their support and encouragement. We would like to thank Editage (www.editage.com) for English language editing.

## Author Contributions

**Conceptualization:** Ryo Oizumi.

**Data curation:** Ryo Oizumi.

**Formal analysis:** Ryo Oizumi, Hisashi Inaba, Youichi Enatsu, Kensaku Kinjo.

**Funding acquisition:** Ryo Oizumi.

**Investigation:** Ryo Oizumi, Takenori Takada.

**Methodology:** Ryo Oizumi, Youichi Enatsu.

**Project administration:** Ryo Oizumi.

**Resources:** Ryo Oizumi, Hisashi Inaba, Takenori Takada.

**Software:** Ryo Oizumi.

**Supervision:** Ryo Oizumi, Hisashi Inaba, Kensaku Kinjo.

**Validation:** Ryo Oizumi, Kensaku Kinjo.

**Visualization:** Ryo Oizumi.

**Writing – original draft:** Ryo Oizumi.

**Writing – review & editing:** Ryo Oizumi, Hisashi Inaba, Takenori Takada, Youichi Enatsu, Kensaku Kinjo.

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
