## [Decision Letter · Decision Letter 0]

17 Jun 2022

PONE-D-22-13284Sensitivity Analysis on the Declining Population in Japan: Effects of Prefecture-Specific Fertility and Interregional MigrationPLOS ONE

Dear Dr. Oizumi,

Thank you for submitting your manuscript to PLOS ONE. After careful consideration, we feel that it has merit but does not fully meet PLOS ONE’s publication criteria as it currently stands. Therefore, we invite you to submit a revised version of the manuscript that addresses the points raised during the review process.

We received review comments from three reviewers. They are all experts in demographic models. Overall, their comments are positive. At this stage, I would like you to focus on improving the clarity of the paper. One of the reviewers raised some concerns about the numerical models. The authors may want to add some description in a case similar questions are raised by the readers, or the level of the confidence in the conclusion can be toned down.

We look forward to receiving your revised manuscript.

Kind regards,

Masami Fujiwara, PhD

Academic Editor

PLOS ONE

Journal Requirements:

3 .Please note that PLOS ONE has specific guidelines on code sharing for submissions in which author-generated code underpins the findings in the manuscript. In these cases, all author-generated code must be made available without restrictions upon publication of the work. Please review our guidelines at https://journals.plos.org/plosone/s/materials-and-software-sharing#loc-sharing-code and ensure that your code is shared in a way that follows best practice and facilitates reproducibility and reuse.

Reviewers' comments:

Reviewer's Responses to Questions

**Comments to the Author**

1. Is the manuscript technically sound, and do the data support the conclusions?

Reviewer #1: Yes

Reviewer #2: Yes

Reviewer #3: Yes

2. Has the statistical analysis been performed appropriately and rigorously? 

Reviewer #1: N/A

Reviewer #2: N/A

Reviewer #3: N/A

3. Have the authors made all data underlying the findings in their manuscript fully available?

Reviewer #1: Yes

Reviewer #2: Yes

Reviewer #3: Yes

4. Is the manuscript presented in an intelligible fashion and written in standard English?

Reviewer #1: Yes

Reviewer #2: Yes

Reviewer #3: Yes

5. Review Comments to the Author

Reviewer #1: Summary:

The authors tackle the important demographic topic of the declining population in Japan and low birth rate. They use a multi-regional Leslie matrix model to quantitatively evaluate the factors of recent population decline, and develop a novel method for representing the reproductive value and stable age distribution for a space-age-structured population. Also of novel interests, the method provides inference on demographic outcomes using genealogies of the migration history of individuals and their ancestors. By combining the new method with classical sensitivity analysis, the authors analyze the effect of prefecture-specific fertility rates and inter-regional migration rates on the population decline in Japan which can help guide policy planning and future research. Below I highlight a few general comments on how the clarity of the paper could be improved, and also provide a few minor grammatical edits. Overall, I think the paper will be of high interest to human demographers across the globe and to policy makers in Japan.

General Comments:

Abstract, lines 11 and 14: Are you referring to the sensitivities of migration/fertility rates to changes in regional traits, OR the sensitivity of population growth rate to changes in migration/fertility for specific types of regions? After reading the Introduction I think you imply the latter, you should reword these lines of the Abstract to make this clear. This issue arises elsewhere in the paper and I might not have caught all of the instances.

Lines below refer to line numbers provided on the far right-side of the PDF that was generated for reviewers.

11-14: This sentence is phrased as if human population decline is always a bad thing. In the near-term, I fully understand the deleterious impact on the economy, health insurance programs, and number of other issues that can affect human well-being in the near-term, but in the moderate-term it might be a great outcome for sustainability and human livelihoods leading up to an eventual transition back to replacement-level fertility (and with momentum of course affecting population dynamics as this occurs). I would suggest a minor rewording along the lines of: … significant measures against population decline, or to optimally plan for it, could be implemented.

17: delete ‘using a stable population model’ because it is not a prerequisite for sensitivity analysis (which can be done in nonlinear systems, stochastic systems, transient dynamics, some of which you refer to in later text). Then for the following sentence lead with ‘For a stable population model, …’

There is nothing to prevent you from using the GLMM acronym for generalized Leslie matrix model, but I will note that it is much more commonly used for Generalized Linear Mixed Model (a statistical model used in many fields), which could create confusion for some readers.

65-75: Excellent! This is a very novel development in demography that could be applied outside of human demography.

90-91: I think these lines are not necessary and could be deleted. As such, the sub-heading on line 89 could potentially be deleted as well.

Eq. 43 (and 3 lines after Eq. 49): Here you use italic ‘m’ to refer to fertility, but elsewhere you use italic ‘f’.

436-481: It is not clear how this section relates to your objectives and findings, and is currently a bit tangential. Though not as common, there are continuous models of population dynamics in animal ecology (especially for invertebrates). In addition, you do not mention one of the main decision points for choosing a continuous vs. discrete model, which is whether the focal organism has birth-flow or birth-pulse reproduction (but of course acknowledging that morality dynamics are always continuous). Moreover, I don’t see a comparison of your discrete-age model to a continuous differential equation…which is why I say this section is tangential. It’s as if you are trying to justify your use of a Leslie model to human demographers, but they are used commonly. If the section is necessary, it requires refocusing of the text.

Minor Grammatical Edits:

Abstract, 5 lines from the bottom: ‘stable age distribution’ instead of ‘stable distribution’

Lines below refer to line numbers provided on the far right-side of the PDF that was generated for reviewers.

6: …varies with each geo-political area.

8-9: …and indicates continuation of the population decline…

20: primarily

78: population growth rate

167, 168, 170: birth rates

185: born

244: Although population growth is sensitive to all entries, those with…

246: Thus, all entries can affect all eigenvalue sensitivities in general

251: …is sufficient for asymptotic inference.

272: expects population growth rate to be more sensitive to elder fertility

528: …the sensitivity of population growth rate to migration ??

This is a great contribution to human demography and application to the demography of Japan!

Dave Koons

Reviewer #2: Demographic transition is a global phenomenon. Declining birthrate and death rate lead to population aging and eventurly population declining. Japan definitely is one of the most impacted countries. Understanding the causes of population declining, including but not limited to birthrate, migration and other environment traits, is important for dealing with the issue. The manuscript studies the demographic dynamics of Japan via a transition matrix model. Authors construct a multi-regional Leslie matrix and analyze the sensitivity of region-specific fertility rates and interagional migration rates. It is an excellent way to utilize a well developed and established model in population ecology into human demographic. Authors well explain how to construct a mega-matrix including multi-regional information which is a useful guideline for similar researches in this spectrum. Authors also provide detailed explanation on how to obtain and understand the matrix related parameters, such as eigenvalues, left and right eigenvectors, and the sensitivity on eigenvalue. It is a well written and clear addressed method paper on mega-Leslie matrix model. This manuscript should be able to provide demographers an extra tool to understanding human population dynamics.

However, I am not sure the results from their numerical analyses, with the 5-year interval vital rates that are only on residence but not migrants, should be used in policy making. The numerical analyses in this manuscript is a great sample exercise to show how the model works. It is somewhat misleading if audience consider the paper provides a comprehensive study on the sensitivity of the declining population in Japan from views of the fertility and migration. As authors state in the discussion, “Eq. (5) assumes that migrants exhibit the same reproductive and death behaviors as people living in each destination prefecture. In reality, however, the life course depends on the length of residence, age, and cultural background of the migrants.” It is not the birthplace setup the vital rates, such as birthrate. People determines the vital rates. The migrants definitely affect the vital rates of a place. A useful model is introduced here, but more detailed data and comprehensive analyses via this model are need to understand the effects of prefecture-specific fertility and interregional migration on the declining population in Japan. Authors need to reword their abstract and discussion in order to make this clear.

Typos:

Line 185: “bornn” → “born”

Line 257: an unknown citation after Leslie matrix model

Line 385: add a space after “)”

Reviewer #3: In this manuscript, the authors use sensitivity analysis of a multiregional matrix model to determine policy interventions that could influence population decline in Japan. They develop closed-form expressions for the eigenvectors of the multiregional projection matrix (lines 118-197) and use them to express sensitivities of population growth with respect to projection matrix entries (lines 237-276). They also derive type-reproduction number, an analogous quantity to R_0 for individual prefectures (lines 198-236). Then they parameterize the model using census data with 5-year intervals (lines 277-331). They determine that migration of females from urban areas to high-fertility rural prefectures would most effectively counter population decline.

I like how the analysis and discussion illuminate the differing effects of population density and fertility, such as in the effects of late-age fertility in urban areas vs. earlier fertility in less-urbanized ones (e.g., lines 564-575). It’s excellent that the authors obtained the data to parameterize the model. The conclusions seem eminently reasonable to me, and I did not spot any errors. One recommendation I would make would be for the authors to search the term “megamatrix” in the ecological literature, as there is precedent for considering the sensitivities/elasticities of multiply-structured populations under that name (e.g., Tuljapurkar, Horvitz, Pascarella 2003 in The American Naturalist, and I turned up at least one more recent empirical paper by Warchola, Crone, Schultz 2017 Journal of Applied Ecology, that is described with that term and could be relevant as it considers management of an endangered butterfly). Although I believe the authors’ statement that general theory is lacking (lines 49-50), biologists have been proceeding by computing the eigenvectors in the usual way from the megamatrix (Eqn 5 here) as for a standard Leslie or Lefkovitch matrix. I would therefore recommend that the authors explicitly state and emphasize what is novel here, in light of that biological literature. One potential candidate for being brought forward in this way might be the type-reproduction number. It seems to be novel and to require the present analysis, but I found that its interest value and/or usefulness is actually not very easy to extract from way it is presented in this manuscript.

Finally, if I’m not mistaken, it appears that migration appears in the model to have immediate effects on fertility. In reality it seems unlikely that merely relocating females would alone change fertility outcomes profoundly, at least during the lifetime of given females, unless occurring early enough. The results regarding migration are entirely sensible outcomes of the model and the analysis; I only question whether they would translate to actionable, effective policy. If not, the analyses’ primary utility might be to highlight how difficult it would be actually to change factors driving population decline.

6. PLOS authors have the option to publish the peer review history of their article (what does this mean?). If published, this will include your full peer review and any attached files.

Reviewer #1: No

Reviewer #2: No

Reviewer #3: No

---

## [Author Response · Author response to Decision Letter 0]

7 Aug 2022

Dear Dr. Fujiwara:

We would like to resubmit the attached manuscript, entitled “Sensitivity Analysis on the Declining Population in Japan: Effects of Prefecture-Specific Fertility and Interregional Migration.” The manuscript ID is PONE-D-22-13284.

The manuscript has been carefully rechecked and appropriate changes have been made in accordance with the reviewers’ suggestions. The major changes made are highlighted in the revised document. The responses to their comments have also been prepared and are attached herewith. In addition, the following financial information regarding the funding for this study was provided:

1. the Japan Society for the Promotion of Science (JSPS) KAKENHI (Grant number 20K14368) 

2. the Ministry of Health, Labour and Welfare (Grant number 20AA2007).

We thank you and the reviewers for your thoughtful suggestions and insights, which have enriched the manuscript and produced a more balanced and better account of the research. We hope that the revised manuscript is now suitable for publication in PLOS ONE.

Thank you for your consideration. I look forward to hearing from you.

Response to reviewers:

To Reviewer #1: 

>The authors tackle the important demographic topic of the declining population in Japan and low birth rate. They use a multi-regional Leslie matrix model to quantitatively evaluate the factors of recent population decline, and develop a novel method for representing the reproductive value and stable age distribution for a space-age-structured population. Also of novel interests, the method provides inference on demographic outcomes using genealogies of the migration history of individuals and their ancestors. By combining the new method with classical sensitivity analysis, the authors analyze the effect of prefecture-specific fertility rates and inter-regional migration rates on the population decline in Japan which can help guide policy planning and future research. Below I highlight a few general comments on how the clarity of the paper could be improved, and also provide a few minor grammatical edits. Overall, I think the paper will be of high interest to human demographers across the globe and to policy makers in Japan.

>General Comments:

Abstract, lines 11 and 14: Are you referring to the sensitivities of migration/fertility rates to changes in regional traits, OR the sensitivity of population growth rate to changes in migration/fertility for specific types of regions? After reading the Introduction I think you imply the latter, you should reword these lines of the Abstract to make this clear. This issue arises elsewhere in the paper and I might not have caught all of the instances.

A.　 We apologize for the use of misleading expressions. In addition to the correction, we have rewritten the sentence in which "sensitivity" is the subject as "sensitivity of population growth rate" or "sensitivity of λ".

>Lines below refer to line numbers provided on the far right-side of the PDF that was generated for reviewers.

>11-14: This sentence is phrased as if human population decline is always a bad thing. In the near-term, I fully understand the deleterious impact on the economy, health insurance programs, and number of other issues that can affect human well-being in the near-term, but in the moderate-term it might be a great outcome for sustainability and human livelihoods leading up to an eventual transition back to replacement-level fertility (and with momentum of course affecting population dynamics as this occurs). I would suggest a minor rewording along the lines of: … significant measures against population decline, or to optimally plan for it, could be implemented.

A. Thank you very much for your suggestion. We agree with your amendment. We changed the words that you suggested, please check the line.　

>17: delete ‘using a stable population model’ because it is not a prerequisite for sensitivity analysis (which can be done in nonlinear systems, stochastic systems, transient dynamics, some of which you refer to in later text). Then for the following sentence lead with ‘For a stable population model, …’

A. Thank you very much for your comment. We changed the sentence that you said, please check the line.　

>There is nothing to prevent you from using the GLMM acronym for generalized Leslie matrix model, but I will note that it is much more commonly used for Generalized Linear Mixed Model (a statistical model used in many fields), which could create confusion for some readers.

A. Thank you very much for your noting. We do not use the GLMM acronym for generalized Leslie matrix model and use its formal name.

>65-75: Excellent! This is a very novel development in demography that could be applied outside of human demography.

A. We are delighted with your compliments and motivated.

>90-91: I think these lines are not necessary and could be deleted. As such, the sub-heading on line 89 could potentially be deleted as well.

A. Thank you very much. I deleted it as you said.

>Eq. 43 (and 3 lines after Eq. 49): Here you use italic ‘m’ to refer to fertility, but elsewhere you use italic ‘f’.

A. Thank you very much. I corrected such mistake in the new manuscript.

>436-481: It is not clear how this section relates to your objectives and findings, and is currently a bit tangential. Though not as common, there are continuous models of population dynamics in animal ecology (especially for invertebrates). In addition, you do not mention one of the main decision points for choosing a continuous vs. discrete model, which is whether the focal organism has birth-flow or birth-pulse reproduction (but of course acknowledging that morality dynamics are always continuous). Moreover, I don’t see a comparison of your discrete-age model to a continuous differential equation…which is why I say this section is tangential. It’s as if you are trying to justify your use of a Leslie model to human demographers, but they are used commonly. If the section is necessary, it requires refocusing of the text.

A.　Thank you for your input. Indeed, this section and sentence seemed useless. It was initially intended to explain the policy differences with the IIASA project, but it may have been an afterthought for the reader. Therefore, I have decided to delete it as you suggested.

>Minor Grammatical Edits:

Abstract, 5 lines from the bottom: ‘stable age distribution’ instead of ‘stable distribution’

Lines below refer to line numbers provided on the far right-side of the PDF that was generated for reviewers.

6: …varies with each geo-political area.

8-9: …and indicates continuation of the population decline…

20: primarily

78: population growth rate

167, 168, 170: birth rates

185: born

244: Although population growth is sensitive to all entries, those with…

246: Thus, all entries can affect all eigenvalue sensitivities in general

251: …is sufficient for asymptotic inference.

272: expects population growth rate to be more sensitive to elder fertility

528: …the sensitivity of population growth rate to migration ??

A. Thank you very much. We have corrected all the expressions you pointed out.

>This is a great contribution to human demography and application to the demography of Japan!

Dave Koons

A. Thank you again for your high evaluation. We will continue to work hard on our research to meet your expectations.

To Reviewer #2: Demographic transition is a global phenomenon. Declining birthrate and death rate lead to population aging and eventurly population declining. Japan definitely is one of the most impacted countries. Understanding the causes of population declining, including but not limited to birthrate, migration and other environment traits, is important for dealing with the issue. The manuscript studies the demographic dynamics of Japan via a transition matrix model. Authors construct a multi-regional Leslie matrix and analyze the sensitivity of region-specific fertility rates and interagional migration rates. It is an excellent way to utilize a well developed and established model in population ecology into human demographic. Authors well explain how to construct a mega-matrix including multi-regional information which is a useful guideline for similar researches in this spectrum. Authors also provide detailed explanation on how to obtain and understand the matrix related parameters, such as eigenvalues, left and right eigenvectors, and the sensitivity on eigenvalue. It is a well written and clear addressed method paper on mega-Leslie matrix model. This manuscript should be able to provide demographers an extra tool to understanding human population dynamics.

However, I am not sure the results from their numerical analyses, with the 5-year interval vital rates that are only on residence but not migrants, should be used in policy making. The numerical analyses in this manuscript is a great sample exercise to show how the model works. It is somewhat misleading if audience consider the paper provides a comprehensive study on the sensitivity of the declining population in Japan from views of the fertility and migration. As authors state in the discussion, “Eq. (5) assumes that migrants exhibit the same reproductive and death behaviors as people living in each destination prefecture. In reality, however, the life course depends on the length of residence, age, and cultural background of the migrants.” It is not the birthplace setup the vital rates, such as birthrate. People determines the vital rates. The migrants definitely affect the vital rates of a place. A useful model is introduced here, but more detailed data and comprehensive analyses via this model are need to understand the effects of prefecture-specific fertility and interregional migration on the declining population in Japan. Authors need to reword their abstract and discussion in order to make this clear.

A. Thank you for your point and feedback. The fertility rate of each individual may indeed depend on their life course. On the other hand, Japanese government statistics do not distinguish between long-term residents and migrants. That means that even in our model, ignoring international population migration, birth rate, death rate, and interregional migration all include the behavior of foreigners and overseas Japanese. (1) As of 2015, the foreign population is about 2% of the total population in Japan. In addition, the Great East Japan Earthquake and the nuclear power plant accident occurred between 2010 and 2015; the Great East Japan Earthquake and the nuclear power plant accident occurred between 2010 and 2015, causing an outflow of the population for about three years. However, the effect of international migration is limited because the sensitivity analysis shows that the structural change between 2010 and 2015 is small. I have added this point (1) to the discussion. In addition, as you mentioned, our study cannot show that the characteristics of births and deaths in each region can ignore the residence duration and the residents' cultural background. Therefore, I added (2) at the back of the discussion, "To make policy decisions, it may be necessary to understand the regional characteristics of fertility and mortality based on the ratio of long-term residents to new residents in each municipality. Was added. I changed the wording in the abstract to "analyze national factors." I hope these changes meet your purpose.

>Typos:

Line 185: “bornn” → “born”

Line 257: an unknown citation after Leslie matrix model

Line 385: add a space after “)”

A. Thank you very much. We have corrected all the expressions you pointed out.

To Reviewer #3: In this manuscript, the authors use sensitivity analysis of a multiregional matrix model to determine policy interventions that could influence population decline in Japan. They develop closed-form expressions for the eigenvectors of the multiregional projection matrix (lines 118-197) and use them to express sensitivities of population growth with respect to projection matrix entries (lines 237-276). They also derive type-reproduction number, an analogous quantity to R_0 for individual prefectures (lines 198-236). Then they parameterize the model using census data with 5-year intervals (lines 277-331). They determine that migration of females from urban areas to high-fertility rural prefectures would most effectively counter population decline.

I like how the analysis and discussion illuminate the differing effects of population density and fertility, such as in the effects of late-age fertility in urban areas vs. earlier fertility in less-urbanized ones (e.g., lines 564-575). It’s excellent that the authors obtained the data to parameterize the model. The conclusions seem eminently reasonable to me, and I did not spot any errors. One recommendation I would make would be for the authors to search the term “megamatrix” in the ecological literature, as there is precedent for considering the sensitivities/elasticities of multiply-structured populations under that name (e.g., Tuljapurkar, Horvitz, Pascarella 2003 in The American Naturalist, and I turned up at least one more recent empirical paper by Warchola, Crone, Schultz 2017 Journal of Applied Ecology, that is described with that term and could be relevant as it considers management of an endangered butterfly). Although I believe the authors’ statement that general theory is lacking (lines 49-50), biologists have been proceeding by computing the eigenvectors in the usual way from the megamatrix (Eqn 5 here) as for a standard Leslie or Lefkovitch matrix. I would therefore recommend that the authors explicitly state and emphasize what is novel here, in light of that biological literature. One potential candidate for being brought forward in this way might be the type-reproduction number. It seems to be novel and to require the present analysis, but I found that its interest value and/or usefulness is actually not very easy to extract from way it is presented in this manuscript.

Finally, if I’m not mistaken, it appears that migration appears in the model to have immediate effects on fertility. In reality it seems unlikely that merely relocating females would alone change fertility outcomes profoundly, at least during the lifetime of given females, unless occurring early enough. The results regarding migration are entirely sensible outcomes of the model and the analysis; I only question whether they would translate to actionable, effective policy. If not, the analyses’ primary utility might be to highlight how difficult it would be actually to change factors driving population decline.

A. Thank you for your suggestion. As far as I can tell from the paper you presented, the eigensystems of the mega-matrices are derived from an approximate theory based on numerical computation. Of course, they are functions of the matrix entries, but they do not represent the functions. Previous analyses of mega-matrices have analyzed life histories by computing sensitivities and elasticities using the numerical values constituting the eigensystems. Our method represents the eigensystems themselves as matrix entries.　We have added this explanation to the discussion along with a citation of the paper you presented. I added the most important novelty of this paper to the argument with the phrase "our theorem extends the relationship between the basic reproduction number and the Euler-Lotka equation in the Leslie matrix to irreducible mega-matrices." I hope these will be to your liking.

Sincerely,

Ryo Oizumi

National Institute of Population and Social Security Research

2-2-3 Uchisaiwai-cho, Chiyoda-ku

Tokyo 100-8916, Japan

Email: ooizumi-ryou@ipss.go.jp

---

## [Editor Report · Decision Letter 1]

16 Aug 2022

Sensitivity Analysis on the Declining Population in Japan: Effects of Prefecture-Specific Fertility and Interregional Migration

PONE-D-22-13284R1

Dear Dr. Oizumi,

We’re pleased to inform you that your manuscript has been judged scientifically suitable for publication and will be formally accepted for publication once it meets all outstanding technical requirements.

Kind regards,

Masami Fujiwara, PhD

Academic Editor

PLOS ONE
---

## [Editor Report · Acceptance letter]

18 Aug 2022

PONE-D-22-13284R1 

Sensitivity Analysis on the Declining Population in Japan: Effects of Prefecture-Specific Fertility and Interregional Migration 

Dear Dr. Oizumi:

I'm pleased to inform you that your manuscript has been deemed suitable for publication in PLOS ONE. Congratulations! Your manuscript is now with our production department. 

Kind regards, 

on behalf of

Dr. Masami Fujiwara 

Academic Editor

PLOS ONE